# Developing the Fitness of Law Enforcement Recruits during Academy Training

**Danny J. Maupin** [1,*] **, Ben Schram** [1,2] **, Elisa F. D. Canetti** [1,2] **, Jay J. Dawes** [2,3] **, Robert Lockie** [4] **and Robin M. Orr** [1,2]

1   Faculty of Health Sciences and Medicine, Bond University, Robina, QLD 4226, Australia;
    bschram@bond.edu.au (B.S.); ecanetti@bond.edu.au (E.F.D.C.); rorr@bond.edu.au (R.M.O.)
2   Tactical Research Unit, Bond University, Robina, QLD 4226, Australia; jay.dawes@okstate.edu
3   Department of Kinesiology, California State University Fullerton, Fullerton, CA 92831, USA
4   School of Kinesiology, Oklahoma State University, Stillwater, OK 74078, USA; rlockie@fullerton.edu
*   Correspondence: dmaupin@bond.edu.au

**Abstract:** Law enforcement is an intermittently physically demanding job, interspersed with long periods of sedentary activity. To prepare for the physical demands of the job, law enforcement agencies enlist recruits into academies with a focus on physical training. Often, academies focus on aerobic-based exercise despite anaerobic fitness being strongly correlated to occupational tasks. The objective of this article is to analyze the changes in the fitness of police recruits during academy training. Initial and final fitness test results, encompassing muscular power, strength, endurance as well as aerobic and anaerobic fitness, were measured to analyze changes in fitness. Dependent *t*-tests showed significant increases ($p < 0.05$) across all fitness tests, with a trend towards larger increases in aerobic and muscle-endurance-based tests. Recruits from this academy tended to have higher fitness results compared to other academies and were either average or below average compared to age-matched standards in the general population. Physical training should persist for recruits beyond the academy to continue to develop fitness throughout their career. Academies should add a focus on muscular strength and power training as these measures relate to occupational tasks, which may better prepare recruits for demands they will be expected to face in the field.

**Keywords:** conditioning; police; anaerobic fitness; physical training

## 1. Introduction

Law enforcement is a physically demanding occupation requiring personnel to repeatedly and, on occasion, immediately shift from periods of sedentary behavior to high-intensity activity [1]. For example, an officer may transition from sitting in a patrol car to maximal running speed to chase and apprehend a suspect [1]. In addition, officers must perform these high-intensity activities while carrying up to 10 kg of additional occupational load [2]. This additional load has a negative impact on task performance, while also increasing injury risk [2–4]. Thus, officers must maintain a sufficient level of fitness to perform these high-intensity activities adequately and safely.

Law enforcement agencies commonly use academies as a means of preparing recruits for a career in law enforcement. This period of training must develop multiple skills and qualities in individuals by not only teaching the necessary skills and procedures for working as a police officer but by preparing recruits for the physical and psychological challenges of working in law enforcement [5–7]. Physical training is a core component of police training at the academy and is used to prepare recruits for the physical nature of a career in law enforcement. Anaerobic fitness, in particular, including power and strength, is positively correlated with occupational tasks in policing such as a victim drag or wall

climb [8,9]. Despite this relationship, academies often focus on aerobic-based (e.g., formation runs) and muscular endurance (e.g., bodyweight exercises) due to large class sizes and limited availability of equipment [10,11].

Developing physical fitness during the academy not only prepares recruits to handle the occupational workload but may benefit personnel throughout their career. Previous research suggests that as officers progress through their careers, they experience a decrease in physical fitness [1]. If academies train recruits to a higher level of fitness upon graduation, they may be more resilient to fitness decline. Improving the fitness of recruits can also lead to officers with a lower risk of injury [12] and improved long-term psychological [13] and physical [14] well-being. Due to the multiple benefits of improving fitness in recruits on job performance and, psychological, and physical health, it is imperative to profile the fitness development in law enforcement academies to ensure it is being trained effectively. Profiling these developments will allow for specific and focused interventions, if necessary, to further improve recruit training. It is important to implement these on a case by case basis as different agencies will often have different fitness requirements [15].

Therefore, the aim of this article is to profile the physical fitness developed by a specific law enforcement academy, and how it relates to occupational task performance. Additionally, results from this study will be compared to other academies as well as the general population to provide context around the recruits' levels of fitness upon graduation.

## 2. Materials and Methods

### 2.1. Subjects

Data were retrospectively collected from 10 academy recruit classes, totaling 715 participants. Of these 715 participants, 604 were male (age = 26.70 ± 5.22 y, height = 175.98 ± 7.37 cm, body mass = 83.16 ± 12.29 kg), 110 were female (age = 26.69 ± 4.64 y, height = 162.63 ± 6.56 cm, body mass = 65.32 ± 12.08 kg), with one participant not disclosing their sex (age = 38 years, height = 162.50 cm, body mass = 58.60 kg). The majority of subjects in this sample were male, which is commonly seen in other studies of comparable populations [1,6,16]. Ethics approval was obtained by the Bond University Human Research Ethics Committee and by the California State Fullerton Institutional Review Board—both under HSR-17-0037.

Recruit training consisted of 36 physical training sessions, with three of these sessions consisting of fitness tests. Training sessions varied across the academy, but often consisted of two to four sessions per week (lasting approximately two hours each). Training sessions often entailed long-distance formation running, bodyweight exercises, and circuit training. In addition to organized physical training sessions, recruits also engaged in various incentive training and defensive tactic sessions. These sessions were organized and supervised by recruit training instructors who had previously undergone a two-week physical training instruction course.

### 2.2. Procedures

#### 2.2.1. PT500

The PT500 is a composite score of six assessments: maximal push-ups, sit-ups, and mountain climbers completed in 120 s; maximal pull-ups; a 201-m run; and a 2.4-km run. The PT500 is an established standard of fitness assessment that has been used historically with the Los Angeles Sheriff's Department (LASD) [7,17]. Recruits completed the assessments in typical physical training attire. The push-ups, sit-ups, and mountain climbers were completed on an outdoor, concrete surface with a partner, who ensured correct techniques and counted the number of repetitions. Pullups were completed on an outdoor pullup bar. The 201-m and 2.4-km runs were completed on an athletic track at the LASD training facility. Recruits completed the runs in groups of 10–15. Specific procedures for

each of the assessments have been published in previous research [7,17] but are described in detail below for reference. The scoring system for each regarding the final PT500 score is likewise detailed.

### 2.2.2. Push-Ups

The maximal number of push-ups a recruit could complete in 120 s was assessed. Recruits started in the standard "up" position, with the body straight, hands positioned shoulder-width apart, and fingers pointed forwards. A water bottle was placed under the recruits' chest to determine the correct depth of the "bottom" position of the push-up. Upon start, LASD staff began timing the 120 s, and recruits flexed their elbows, lowering themselves until their chest touched the water bottle. Recruits then extended their elbows, returning to the start position. This technique was completed as many times as possible in the 120 s. Recruits were awarded one point per push-up completed, with a max score of 50.

### 2.2.3. Sit-Ups

To test abdominal muscular endurance, the maximum number of sit-ups that could be completed in 120 s was evaluated. Recruits laid on their backs, with knees flexed to 90 degrees, feet flat on the ground, and hands cupped behind ears. Each recruit had a partner holding their feet to the ground during the test. Upon start, training staff began timing. Recruits raised their shoulders from the ground until their elbows touched their knees, while keeping feet flat on the ground and hands cupped behind their ears. The recruits would then lower themselves down until their shoulder blades contacted the ground. This technique was performed as many times as possible in 120 s. For the first 50 repetitions, recruits were given one point per repetition, while for the last 25 repetitions, they were given two points per repetition.

### 2.2.4. Mountain Climbers (MCs)

Another assessment of muscular endurance, MCs involve isometric work in the trunk and upper limb musculature with dynamic movement occurring in the hip and knee joints. Recruits started in the standard "up" position of a push-up and maintained this position with arms extended throughout the test. Maintaining a neutral spine, recruits alternated flexing the hip and knee for each leg, bringing the knee close to the chest and foot underneath the body with each repetition. Recruits began at the start command, with staff timing the 120 s. The first 40 mountain climbers completed count as one point each, while the last 20 were given three points each.

### 2.2.5. Pull-Ups

The pull-up test provides a second measurement of upper body endurance. The recruits' start position involved hanging on the bar in a vertical position, hands shoulder-width apart, and using a pronated grip. While maintaining a vertical body alignment, recruits pulled themselves upwards until their chin cleared the bar. Recruits would then lower themselves until their arms were fully extended. This technique was continued until the recruit could no longer get their chin over the bar. Each repetition counted as three points with a maximum score of 60 points.

### 2.2.6. 201-m Run

The 201-m run was conducted on a running track and timed by staff members with a stopwatch (Professional Digital Stopwatch Timer, LuckyStone Huron, OH, USA). Upon the start command, training staff started timing and the recruits ran as quickly as possible until they passed the distance marker. Run time was recorded for each recruit to the nearest 0.1 s.

2.2.7. 2.40-km Run

For the 2.40-km run assessment, recruits were required to complete six laps as quickly as possible around a 400-m athletics track at the LASD training facility. Run time was recorded for each recruit on a handheld stopwatch (Professional Digital Stopwatch Timer, LuckyStone Huron, OH, USA) to the nearest 0.1 s.

2.2.8. Work Sample Test Battery (WSTB)

The WSTB is a California-mandated group of tests each law enforcement agency must complete. Recruits must obtain a minimum score of 384 to graduate from the academy where points are awarded relative to the completion time of each task [18]. As with the PT500, this assessment has been previously been described in the literature [7,17].

2.2.9. 99-Yard. Obstacle Course (99OC)

Simulating a foot pursuit, recruits were instructed to complete the 99-yard (90.53 m) course as quickly as possible while remaining on the concrete track. During this run, recruits must also clear three 0.15 × 0.15-m curbs and one 0.86-m high obstacle.

2.2.10. Body Drag (BD)

The body drag assessment is a measure of lower limb power and requires recruits to drag a 74.84-kg dummy for 9.75 m. Initially, recruits were required to pick up the dummy by wrapping their arms underneath the arms of the dummy and extending their hips and knees. Timing was initiated as soon as the recruit began dragging the dummy. Recruits dragged the dummy by walking backwards over the complete 9.75 m at which point timing was stopped. Time was recorded to the nearest 0.1 s.

2.2.11. Chain Link Fence Climb (CLF)

Recruits began 4.57 m away from the fence. Upon starting the test, recruits were required to run up to the fence using whatever technique they choose, without using the side supports to assist their climb. Recruits were given two attempts to scale the fence. Once the fence was cleared, recruits were required to land, and then run 22.86 m, as quickly as possible, to complete the test. Staff measured the time to complete the task using a handheld stopwatch (Professional Digital Stopwatch Timer, LuckyStone Huron, OH, USA). Time was recorded to the nearest 0.1 s.

2.2.12. Solid Wall Fence Climb (SW)

As per the CLF, recruits ran 4.57 m before clearing the fence with any technique and the running 22.86 m upon clearance. The only difference between the two tests was the type of fence that needed to be cleared, with this test utilizing a solid wall instead of a chain link fence. Time was recorded to the nearest 0.1 s using a handheld stopwatch (Professional Digital Stopwatch Timer, LuckyStone Huron, OH, USA).

2.2.13. 500-Yard. Run (500R)

LASD staff marked 500 yards (457.20 m) on an athletics track. Recruits were instructed to run this distance as quickly as possible with training staff standing at the finish line timing each recruit to the nearest 0.1 s using a handheld stopwatch (Professional Digital Stopwatch Timer, LuckyStone Huron, OH, USA).

2.2.14. 20-m Multi-Stage Fitness Test (MSFT)

An MSFT test was completed independently of the PT500 and WSTB. Standard procedures were adopted for the MSFT, with recruits required to run back and forth between two lines 20 m apart. Running speed was standardized by pre-recorded auditory cues played from an iPad handheld device

(Apple Inc., Cupertino, CA, USA) connected to a portable speaker (ION Block Rocker, Cumberland, RI, USA) via Bluetooth. The speaker was located in the center of the running area so each recruit could clearly hear the auditory cues but positioned in a way as not to interfere with the recruits' running. The test was stopped when the recruit was unable to reach markers twice in a row during the allotted time as indicated by the auditory cues, or voluntarily stopped running. Scores were recorded according to the final stage the recruit was able to achieve, and then used to calculate the total number of completed shuttles. $VO_2Max$(mL/kg/min) was estimated for each recruit based on the equation by Ramsbottom et al. [19]. This MSFT has previously been validated for use in police populations [19].

### 2.2.15. Medicine Ball Toss (MBT)

Completed independent of the PT500 and WSTB was a medicine ball toss. This test was used to measure upper-body power. Recruits sat on the ground, with head, shoulders, and lower back touching against a concrete wall. The recruits then tossed a 2-kg medicine ball (Champion Barbell, Irving, TX, USA), lightly dusted with chalk, as far as possible using a two-handed chest past. A standard tape measure was used to measure the perpendicular distance from the wall to the closest chalk mark made by the ball landing. Two trials were completed with a recovery time ranging from 30 to 60 s. Results were recorded to the nearest 0.01 m with the farthest of the two trials being recorded. This procedure has previously been used as a measure of fitness in police recruit populations [20].

### 2.2.16. 75-Yard Pursuit Run (75PR)

Lastly, a 75PR (68.58 m) was also performed. This test consisted of a recruit completing five linear sprints about a square grid with sides measures 12.10 m, while also completing four 45-degree direction changes across the grid. Recruits were also required to step over three barriers (2.44 m long and 0.15 m high) during three of the five sprints. Time was recorded using a handheld stopwatch that began on initiation of movement and ended with the recruit crossing the finish line, measured to the nearest 0.1 s (see Figure 1). The 75PR has previously been used as a measure of fitness and occupational task performance in recruit populations [20].

### 2.3. Statistical Analyses

Data were uploaded into Microsoft Excel version 16.0 to be cleaned. Data were only retained if there was a recording on both testing dates. Statistical analysis was completed using the R Studio Statistical Software version 1.2 (Auckland, NZ). A paired *t*-test was completed using initial and final scores of each of the above tests to determine the overall change in fitness during the academy. Significance was set at $p < 0.05$ a priori. Effect sizes (*d*) for between-group comparisons were calculated for each fitness test by dividing the difference between the means by the pooled standard deviation (SD) [21]. The interpretation of effect sizes is based on Hopkins [22] where values less than 0.20 are considered a trivial effect; 0.20 to 0.60, a small effect; 0.60 to 1.20, a moderate effect; 1.20 to 2.00, a large effect, 2.00 to 4.00, a very large effect; and greater than 4.00, an extremely large effect. Using results from the final fitness tests, means were calculated to build a fitness profile of recruits graduating from the academy. Means were calculated overall and between females and males and then further separated into age ranges to allow comparison to other studies and to general population fitness norms described by the American College of Sports Medicine (ACSM) [23]. SeparationThe separation between age and gender was performed as previous research has shown differences in fitness exist between genders and age ranges in comparable populations [1,16]. The participant who did not disclose their sex was not included in this analysis, but was kept in the overall analysis. To further compare the age ranges, an ANOVA (analysis of variance) test was performed for the males, with Levene's test used to ensure appropriate homogeneity of variance. The effect size for the ANOVA was calculated using $\omega^2$, calculated by subtracting the product of degrees of freedom and regression mean square from the mean sum of squares, and dividing this by the total sum of squares and mean regression square, with the values of 0.01, 0.06, and 0.14 representing small, medium, and large effects respectively.

As they did not declare their ages [24]. 11 males were not included in the analysis of age. Due to not declaring their age. Previous research in comparable populations hashave used similar methods to assess differences in fitness across age and gender [1]. A post hoc analysis was then performed to identify which age ranges significantly differed. As post hoc analyses tend to perform poorly when sample sizes are not equal, a robust post hoc test was performed for all variables even if Levene's test was non-significant [25]. The robust post hoc test utilized involved trimmed means and bootstrapping as proposed by Wilcox [26]. As little or no data were present for the over 40 range for females, analysis was completed comparing the 20–29 age range to the 30–39 age range. This analysis was completed by using the Wilcoxon rank-sum test due to the non-normality present in the various fitness measures of the two groups. Effect size, $r$, was calculated by dividing the $z$-score by the square root of the sample size. Interpretations of the effect size were 0.10, 0.30, and 0.50 for a small, medium, and large effects, respectively effect [27].

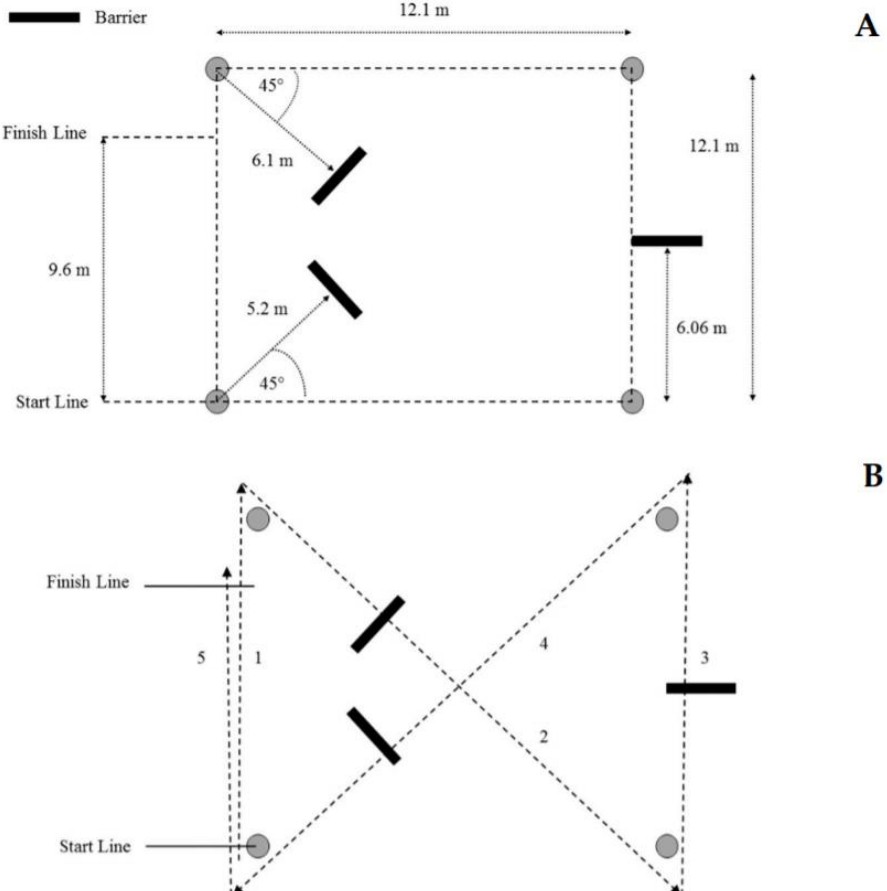

**Figure 1.** Diagram of 75-Yard Pursuit Run. Label A refers to the five linear sprints. Label B refers to four 45-degree direction changes.

A post hoc power analysis was conducted using the G*Power statistical program version 3.1 (Dusseldorf, GER). A post hoc analysis was utilized as this data were retrospectively provided and the sample size could not be changed. Thus, the significance level was set at 0.80 a priori.

## 3. Results

Detailed data describing the mean difference, test statistic, effect size, and power for each outcome are provided in Table 1. Results from the dependent *t*-tests show significant improvements in all fitness measures from initial to final testing, with effect sizes ranging from trivial to moderate. Sufficient

power was present in all tests with the exception of the 99OC (0.52), signifying the sample size was large enough for this analysis.

**Table 1.** Change in Fitness Over the Course of Academy.

| Test | Mean Difference (95% CI) | *p*-Value | Test Statistic | Cohen's *d* | Interpretation | Power |
|---|---|---|---|---|---|---|
| PT500 pts | 80.20 (75.59–84.81) | <0.001 | 34.14 | 0.99 | Moderate | 1.00 |
| Sit-ups reps | 8.79 (7.92–9.64) | <0.001 | 20.04 | 0.66 | Moderate | 1.00 |
| MC reps | 9.75 (8.82–10.69) | <0.001 | 20.50 | 0.87 | Moderate | 1.00 |
| Pull-ups reps | 4.81 (4.26–5.36) | <0.001 | 17.18 | 0.66 | Moderate | 1.00 |
| Push-ups reps | 5.83 (5.03–6.62) | <0.001 | 14.38 | 0.52 | Small | 1.00 |
| 2.4-km run s | −52.97 (−48.09−−57.84) | <0.001 | −21.31 | −0.68 | Moderate | 1.00 |
| 201-m run s | −3.02 (−2.72−−3.31) | <0.001 | −19.91 | −0.66 | Moderate | 1.00 |
| 99OC s | −0.85 (−0.017−−1.69) | 0.045 | −2.01 | −0.12 | Trivial | 0.52 |
| BD s | −0.76 (−0.48−−1.04) | <0.001 | −5.28 | −0.29 | −Small | 1.00 |
| CLF s | −0.40 (−0.21−−0.59) | <0.001 | −4.12 | −0.18 | Trivial | 0.98 |
| SW s | −0.76 (−0.49−−1.03) | <0.001 | −5.59 | −0.27 | Small | 1.00 |
| 500R s | −3.69 (−3.05−−4.35) | <0.001 | −11.22 | −0.35 | Small | 1.00 |
| WSTB pts | 25.99 (19.64–32.33) | <0.001 | 8.05 | 0.30 | Small | 1.00 |
| MSFT shuttles | 21.84 (19.72–23.95) | <0.001 | 20.30 | 1.12 | Moderate | 1.00 |
| VO$_2$max mL/kg/min | 6.05 (5.46–6.65) | <0.001 | 20.00 | 1.03 | Moderate | 1.00 |
| MBT m | 0.34 (0.20–0.48) | <0.001 | 4.75 | 0.23 | Small | 1.00 |
| 75PR s | 0.32 (0.21–0.45) | <0.001 | 5.52 | 0.26 | Small | 1.00 |

Key—MC: mountain climber, 99OC: 99-yard obstacle course, BD: body drag, CLF: chain link fence climb, SW: solid wall climb, 500R: 500-yard run, WSTB: Work Sample Test Battery, MSFT: 20-m multi-stage fitness test, MBT: medicine ball toss, 75PR: 75-yard pursuit run.

Table 2 shows the fitness profiles of recruits at the end of the academy both overall and separated by sex. Males tended to have better scores in nearly all fitness tests, with the exception of MCs. Results of an independent *t*-test analysis (Table 3) confirm this trend with males showing significantly higher scores on all tests except for MCs. All tests also demonstrated appropriate power except for MCs and sit-ups. These results also show larger effect sizes in the tests comprising the WSTB, suggesting that the difference between male and female recruits is even greater in tests of anaerobic fitness.

**Table 2.** Fitness Profile of Recruits at End of Academy.

| Test | Mean ± 95% CI | Sample Size | Female Mean ± 95% CI | Sample Size | Male Mean ± 95% CI | Sample Size |
|---|---|---|---|---|---|---|
| PT500 pts | 411.42 ± 5.18 | 715 | 347.73 ± 10.93 | 110 | 423.18 ± 5.29 | 604 |
| Sit-ups reps | 65.73 ± 0.86 | 715 | 63.54 ± 2.37 | 110 | 66.14 ± 0.92 | 604 |
| MC reps | 58.72 ± 0.55 | 713 | 59.52 ± 1.21 | 110 | 58.57 ± 0.62 | 602 |
| Pull-ups reps | 15.18 ± 0.63 | 635 | 7.64 ± 2.44 | 55 | 15.90 ± 0.62 | 580 |
| Push-ups reps | 52.90 ± 0.75 | 714 | 47.06 ± 1.45 | 110 | 53.98 ± 0.82 | 603 |
| 2.4-km s | 661.21 ± 5.28 | 711 | 701.17 ± 15.47 | 110 | 653.79 ± 5.39 | 600 |
| 201-m s | 31.47 ± 0.30 | 710 | 35.85 ± 0.70 | 110 | 30.64 ± 0.29 | 599 |
| 99OC s | 18.70 ± 0.15 | 453 | 20.33 ± 0.43 | 61 | 18.44 ± 0.15 | 391 |
| BD s | 5.32 ± 0.20 | 475 | 7.11 ± 0.49 | 71 | 5.00 ± 0.21 | 403 |
| CLF s | 7.99 ± 0.12 | 477 | 9.49 ± 0.32 | 72 | 7.72 ± 0.11 | 404 |
| SW s | 7.74 ± 0.14 | 450 | 9.87 ± 0.50 | 52 | 7.45 ± 0.13 | 397 |
| 500R s | 89.27 ± 0.78 | 463 | 99.41 ± 1.61 | 70 | 87.42 ± 0.74 | 392 |
| WSTB pts | 529.24 ± 5.39 | 381 | 455.50 ± 14.32 | 54 | 541.62 ± 4.67 | 326 |
| MSFT shuttles | 70.46 ± 2.25 | 365 | 58.38 ± 4.53 | 58 | 72.74 ± 2.46 | 307 |
| VO$_2$max mL/kg/min | 40.06 ± 0.65 | 365 | 36.41 ± 1.30 | 58 | 40.75 ± 0.71 | 307 |
| MBT cm | 6.31 ± 0.14 | 386 | 4.44 ± 0.15 | 61 | 6.66 ± 0.13 | 325 |
| 75PR s | 17.30 ± 0.13 | 381 | 18.60 ± 0.30 | 59 | 17.06 ± 0.13 | 322 |

Key—MC: mountain climber, 99OC: 99-yard obstacle course, BD: body drag, CLF: chain link fence climb, SW: solid wall climb, 500R: 500-yard run, WSTB: Work Sample Test Battery, MSFT: 20-m multi-stage fitness test, MBT: medicine ball toss, 75PR: 75-yard pursuit run.

Tables 4 and 5 show female and male fitness separated by age, respectively. There appears to be a trend across both sexes of decreasing fitness results as age increases, though sample size for females in the upper ranges is substantially smaller.

**Table 3.** Independent *t*-test Results of Male and Female Recruit Fitness at Academy End.

| Test | Mean Difference (95% CI) | *p*-Value | Test Statistic | Cohen's *d* | Interpretation | Power |
|---|---|---|---|---|---|---|
| PT500 pts | 75.45 (65.33–87.57) | <0.001 | 12.29 | 1.16 | Moderate | 1.00 |
| Sit-ups reps | 2.60 (0.06–5.14) | 0.04 | 2.03 | 0.22 | Small | 0.57 |
| MC reps | 0.94 (−2.30–0.41) | 0.18 | −1.38 | −0.13 | Trivial | 0.24 |
| Pull-ups reps | 8.26 (5.75–10.77) | <0.001 | 6.57 | 1.07 | Moderate | 1.00 |
| Push-ups reps | 6.92 (5.26–8.58) | <0.001 | 8.22 | 0.70 | Moderate | 1.00 |
| 2.4-km run s | −47.38 (−31.01−−63.74) | <0.001 | −5.73 | −0.68 | Moderate | 1.00 |
| 201-m run s | −5.21 (−4.45−−5.98) | <0.001 | −13.54 | −1.43 | Large | 1.00 |
| 99OC s | −1.88 (−1.43−−2.34) | <0.001 | −8.27 | −1.26 | Large | 1.00 |
| BD s | −2.11 (−1.58−−2.64) | <0.001 | −7.87 | −1.01 | Moderate | 1.00 |
| CLF s | −1.77 (−1.44−−2.11) | <0.001 | −10.54 | −1.54 | Large | 1.00 |
| SW s | −2.42 (-1.90−−2.94) | <0.001 | −9.31 | −1.78 | Large | 1.00 |
| 500R s | −11.99 (−10.22−−13.76) | <0.001 | −13.45 | −1.63 | Large | 1.00 |
| WSTB pts | 86.12 (71.10–101.15) | <0.001 | 11.45 | 1.94 | Large | 1.00 |
| MSFT shuttles | 14.36 (9.23–19.50) | <0.001 | 5.55 | 0.68 | Moderate | 0.99 |
| VO$_2$max mL/kg/min | 4.34 (2.87–5.81) | <0.001 | 5.86 | 0.71 | Moderate | 0.99 |
| MBT m | 2.23 (2.03–2.42) | <0.001 | 22.53 | 2.03 | Very Large | 1.00 |
| 75PR s | −1.54 (−1.21−−1.87) | <0.001 | −9.28 | −1.29 | Large | 1.00 |

Key—MC: mountain climber, 99OC: 99-yard obstacle course, BD: body drag, CLF: chain link fence climb, SW: solid wall climb, 500R: 500-yard run, WSTB: Work Sample Test Battery, MSFT: 20-m multi-stage fitness test, MBT: medicine ball toss, 75PR: 75-yard pursuit run.

**Table 4.** Female Fitness Profile Separated by Age.

| Test | 20–29 Years | Sample Size | 30–39 Years | Sample Size | >40 Years | Sample Size |
|---|---|---|---|---|---|---|
| PT500 pts | 347.93 ± 12.61 | 84 | 350.54 ± 24.49 | 24 | 305.50 ± 57.28 | 2 |
| Sit-ups reps | 63.33 ± 2.75 | 84 | 64.04 ± 5.36 | 24 | 66.00 ± 7.07 | 2 |
| MC reps | 59.70 ± 1.41 | 84 | 59.04 ± 2.65 | 24 | 57.50 ± 3.54 | 2 |
| Pull-ups reps | 8.14 ± 2.95 | 44 | 5.64 ± 3.70 | 11 | N/A | 0 |
| Push-ups reps | 46.50 ± 1.68 | 84 | 49.42 ± 2.93 | 24 | 42.50 ± 10.61 | 2 |
| 2.4-km s | 706.52 ± 13.17 | 84 | 676.62 ± 55.45 | 24 | 771.00 ± 41.01 | 2 |
| 201-m s | 35.81 ± 0.79 | 84 | 35.62 ± 1.65 | 24 | 40.50 ± 2.12 | 2 |
| 99OC s | 20.28 ± 0.48 | 47 | 20.49 ± 1.11 | 14 | N/A | 0 |
| BD s | 6.91 ± 0.53 | 54 | 7.74 ± 1.23 | 17 | N/A | 0 |
| CLF s | 9.48 ± 0.39 | 55 | 9.54 ± 0.53 | 17 | N/A | 0 |
| SW s | 9.68 ± 0.58 | 37 | 10.34 ± 1.07 | 15 | N/A | 0 |
| 500R s | 99.53 ± 1.95 | 53 | 99.06 ± 3.02 | 17 | N/A | 0 |
| WSTB pts | 455.23 ± 17.43 | 40 | 456.29 ± 27.74 | 14 | N/A | 0 |
| MSFT shuttles | 59.78 ± 5.30 | 45 | 55.00 ± 9.61 | 12 | 36 | 1 |
| VO$_2$max mL/kg/min | 36.77 ± 1.54 | 45 | 35.42 ± 2.70 | 12 | 31.80 | 1 |
| MBT m | 4.47 ± 0.18 | 47 | 4.36 ± 0.31 | 13 | 3.90 | 1 |
| 75PR s | 18.32 ± 0.30 | 45 | 19.52 ± 0.77 | 13 | 19.36 | 1 |

Key—MC: mountain climber, 99OC: 99-yard obstacle course, BD: body drag, CLF: chain link fence climb, SW: solid wall climb, 500R: 500-yard run, WSTB: Work Sample Test Battery, MSFT: 20-m multi-stage fitness test, MBT: medicine ball toss, 75PR: 75-yard pursuit run.

For males, there is slight statistical support for this trend with results from ANOVA showing significant differences in 11 fitness tests (PT500, sit-ups, push-ups, 2.4-km run, 220-m run, BD, CLF, SW, 500R, WSTB, and 75PR) with a small effect size (Table 6). Post-hoc analysis (Table 7) shows that, with the exceptions of sit-ups, CLF, SW, 500R, and WSTB, significant differences existed between the 20–29 and 30–39 age ranges. The 20–29 age group also significantly outperformed the 30–39 group on all these assessments except for push-ups and MCs. The 20–29 age significantly outperformed the over 40 age group on all assessments with the exception of BD. Lastly, significant differences existed between the 30–39 age group and the 40–49 age group in push-ups, 201-m run, CLF, SW, 500R, and WSTB with the 30–39 age group performing better across all tests.

**Table 5.** Male Fitness Profile Separated by Age.

| Test | 20–29 Years | Sample Size | 30–39 Years | Sample Size | >40 Years | Sample Size |
|---|---|---|---|---|---|---|
| PT500 pts | 427.99 ± 5.91 | 466 | 408.02 ± 12.81 | 104 | 387.61 ± 81.05 | 23 |
| Sit-ups reps | 66.69 ± 1.05 | 466 | 64.59 ± 2.27 | 104 | 61.65 ± 10.46 | 23 |
| MC reps | 58.62 ± 0.70 | 465 | 59.14 ± 1.50 | 103 | 55.96 ± 8.68 | 23 |
| Pull-ups reps | 16.12 ± 1.05 | 451 | 15.53 ± 1.73 | 96 | 13.64 ± 9.29 | 22 |
| Push-ups reps | 53.79 ± 0.93 | 465 | 56.12 ± 2.26 | 104 | 49.35 ± 1.43 | 23 |
| 2.4-km s | 649.53 ± 6.24 | 465 | 668.48 ± 11.83 | 101 | 687.17 ± 60.63 | 23 |
| 201-m s | 30.31 ± 0.31 | 463 | 31.56 ± 0.91 | 102 | 33.96 ± 3.24 | 23 |
| 99OC s | 18.42 ± 0.17 | 303 | 18.47 ± 0.37 | 68 | 19.56 ± 0.99 | 11 |
| BD s | 4.86 ± 0.11 | 310 | 5.53 ± 1.05 | 72 | 5.34 ± 0.98 | 11 |
| CLF s | 7.42 ± 0.12 | 312 | 7.80 ± 0.26 | 71 | 8.59 ± 0.94 | 11 |
| SW s | 7.42 ± 0.15 | 305 | 7.39 ± 0.27 | 71 | 8.37 ± 1.32 | 11 |
| 500R s | 86.82 ± 0.84 | 301 | 89.26 ± 1.88 | 70 | 93.73 ± 6.20 | 11 |
| WSTB pts | 543.54 ± 5.31 | 250 | 540.29 ± 11.52 | 58 | 496.89 ± 35.60 | 9 |
| MSFT shuttles | 73.93 ± 2.67 | 238 | 68.72 ± 6.32 | 61 | 68.00 ± 27.38 | 8 |
| VO$_2$max mL/kg/min | 41.08 ± 0.79 | 238 | 39.76 ± 1.71 | 61 | 39.23 ± 7.78 | 8 |
| MBT m | 6.70 ± 0.15 | 254 | 6.61 ± 0.24 | 61 | 5.99 ± 0.76 | 10 |
| 75PR s | 16.95 ± 0.14 | 252 | 17.34 ± 0.34 | 61 | 18.17 ± 1.79 | 9 |

Key—MC: mountain climber, 99OC: 99-yard obstacle course, BD: body drag, CLF: chain link fence climb, SW: solid wall climb, 500R: 500-yard run, WSTB: Work Sample Test Battery, MSFT: 20-m multi-stage fitness test, MBT: medicine ball toss, 75PR: 75-yard pursuit run.

**Table 6.** ANOVA Results of Male Fitness Separated by Age.

| Test | F | *p*-Value | $\omega^2$ | Interpretation |
|---|---|---|---|---|
| PT500 | 7.37 | 0.001 | 0.02 | Small |
| Sit-ups | 3.21 | 0.04 | 0.01 | Small |
| MCs | 1.59 | 0.21 | 0.002 | Minimal |
| Pull-ups | 1.26 | 0.29 | <0.001 | Small |
| Push-ups | 4.63 | 0.01 | 0.01 | Small |
| 2.4-km run | 6.36 | 0.002 | 0.02 | Small |
| 201-m run | 15.25 | <0.001 | 0.05 | Small |
| 99OC | 2.27 | 0.11 | 0.007 | Minimal |
| BD | 3.04 | 0.05 | 0.01 | Small |
| CLF | 3.88 | 0.02 | 0.01 | Minimal |
| SW | 3.017 | 0.05 | 0.01 | Small |
| 500R | 7.06 | 0.001 | 0.03 | Small |
| WSTB | 5.23 | 0.01 | 0.03 | Small |
| MSFT | 1.57 | 0.21 | 0.004 | Minimal |
| VO$_2$max | 1.46 | 0.23 | 0.003 | Minimal |
| MBT | 1.84 | 0.16 | 0.01 | Small |
| 75PR | 6.65 | 0.001 | 0.03 | Small |

Key—MC: mountain climber, 99OC: 99-yard obstacle course, BD: body drag, CLF: chain link fence climb, SW: solid wall climb, 500R: 500-yard run, WSTB: Work Sample Test Battery, MSFT: 20-m multi-stage fitness test, MBT: medicine ball toss, 75PR: 75-yard pursuit run.

**Table 7.** Post-hoc Analysis of Male Fitness Separated by Age.

| Comparison | PT500 | Sit-ups | Push-ups | 2.4-km run | 201-m run | BD | CLF | SW | 500R | WSTB | 75PR |
|---|---|---|---|---|---|---|---|---|---|---|---|
| 20–29 vs. 30–39 y | 0.002 | 0.08 | 0.04 | 0.003 | 0.001 | 0.01 | 0.11 | 0.85 | 0.07 | 0.3 | 0.03 |
| 20–29 vs. >40 y | 0.005 | 0.01 | 0.01 | 0.01 | <0.001 | 0.11 | <0.001 | 0.002 | 0.01 | <0.001 | 0.02 |
| 30–39 vs. >40 y | 0.12 | 0.13 | <0.001 | 0.22 | 0.001 | 0.53 | 0.01 | 0.003 | 0.04 | <0.001 | 0.09 |

Key—BD: body drag, CLF: chain link fence climb, SW: solid wall climb, 500R: 500-yard run, WSTB: Work Sample Test Battery, 75PR: 75-yard pursuit run.

For comparisons between females of age ranges 20–29 and 30–39, results of a Wilcoxon rank-sum test (Table 8) showed significant differences existing only for the 75PR, with the 20–29 age group performing better.

**Table 8.** Wilcoxon Rank Sum Results Female Fitness Age 20–29 and 30–39.

| Test | *p*-Value | *r* | Interpretation | Power |
|------|-----------|-----|----------------|-------|
| PT500 | 0.52 | −0.06 | Minimal | 0.07 |
| Sit-ups | 0.86 | −0.02 | Minimal | 0.08 |
| MCs | 0.59 | −0.05 | Minimal | 0.11 |
| Pull-ups | 0.65 | −0.06 | Minimal | 0.23 |
| Push-ups | 0.30 | −0.10 | Minimal | 0.51 |
| 2.4-km run | 0.55 | −0.06 | Minimal | 0.34 |
| 201-m run | 0.96 | −0.004 | Minimal | 0.08 |
| 99OC | 0.87 | −0.02 | Minimal | 0.10 |
| BD | 0.12 | −0.18 | Minimal | 0.38 |
| CLF | 0.43 | −0.09 | Minimal | 0.07 |
| SW | 0.22 | −0.17 | Minimal | 0.30 |
| 500R | 0.83 | −0.02 | Minimal | 0.08 |
| WSTB | 0.90 | −0.02 | Minimal | 0.06 |
| MSFT | 0.39 | −0.11 | Minimal | 0.22 |
| VO$_2$max | 0.54 | −0.08 | Minimal | 0.21 |
| MBT | 0.46 | −0.09 | Minimal | 0.15 |
| 75PR | 0.003 | −0.39 | Medium | 0.94 |

Key—MCs: mountain climbers; 99OC: 99-yard obstacle course; BD: body drag; CLF: chain link fence climb; SW: solid wall climb; 500R: 500-yard run; WSTB: Work Sample Test Battery; MSFT: 20-m multi-stage fitness test; MBT: medicine ball toss; 75PR: 75-yard pursuit run.

## 4. Discussion

The aims of this study were to assess whether recruits were able to improve their fitness levels over the course of the academy, and to identify if occupation-specific tasks improved over the same timeframe. The results show that recruits were able to significantly increase their performance in almost all domains, with the exceptions being CLF and SW.

Almost all tests related to the PT500 showed a moderate effect size between initial and final tests, with the exception being push-ups, which resulted in a small effect size. MSFT and VO$_2$max also demonstrated a moderate effect size, with MBT and 75PR showing a small effect size. In contrast, tests forming the WSTB ranged from trivial (990C, CLF, SW) to small (BD, 500R, and WSTB) effect sizes. When analyzing the types of tests, it can be seen that the PT500 tends to assess muscular endurance and aerobic capacity (with the possible exception of the 201-m run which only had a small effect size), while the WSTB tends to assess muscular strength and power. (see Supplementary Materials for full breakdown). The MSFT and VO$_2$ are also measures of aerobic capacity, with the MBT being a measure of upper body power, and 75PR relying more on lower body muscular power than endurance. This trend of a greater increase in aerobic fitness and muscular endurance can be seen in other recruit populations. For example, Cocke et al. [6] found greater increases in push-up and sit-up performance compared to one repetition maximum bench press and vertical jump, despite employing a variety of training programs.

While there did appear to be a slight trend towards higher fitness in younger age groups, this trend had minimal statistical support, with age only accounting for a small amount of variance among all fitness tests. The results here follow a similar trend presented by Dawes et al. [1] which showed decreased fitness in older age groups in police officers. However, given the small effect sizes, there may be other variables, such as prior training history, explaining the variance in the fitness results. Further research is needed to account for this and may lead to more sustained fitness over officers' careers as they age.

When comparing the final results of aerobic fitness to other law enforcement agencies, it can be seen that recruits from this academy were able to complete significantly more MSFT shuttles (70.46 ± 2.25 shuttles) compared to recruits from a different agency (61.20 ± 16.98 shuttles) [28]. Female (701.17 ± 15.47 s) and male recruits (653.79 ± 5.39 s) in this academy were also able to complete a 2.4-km run faster than recruits at another academy (female = 741.00 s, male = 660.60 s) [16]. These results suggest that graduates from this academy possess higher levels of aerobic fitness when compared to other police academies. However, when compared to the general population, the VO$_2$max scores for males age 20–29 (41.08 ± 0.79 mL/kg/min) would be considered poor, while females in this age

range (36.77 ± 1.54) would be considered fair according to the ACSM. For the age ranges 30–39, males (39.76 ± 1.71 mL/kg/min) would be considered fair while females (35.43 ± 2.70 mL/kg/min) would be considered good. Finally, males over 40 (36.98 ± 5.56 mL/kg/min) would be considered fair, while females of this age were not able to be compared due to low numbers.

A similar trend can be seen in muscular endurance with recruits graduating from this academy being able to perform more push-ups (52.90 ± 0.75 vs. 48.67 ± 11.87 reps) and sit-ups (65.73 ± 0.86 vs. 44.17 ± 5.91 reps) compared to another academy [28]. When broken down into sexes, it can be seen that females in this academy were able to perform slightly fewer push-ups (47.06 ± 1.45 vs. 51.11 ± 12.75 reps) and more sit-ups (63.54 ± 2.37 vs. 46.83 ± 6.82 reps), while males performed fewer push-ups (53.98 ± 0.82 vs. 70.24 ± 12.27 reps) and more sit-ups (66.14 ± 0.92 vs. 47.92 ± 5.65 reps) [16]. However, it should be noted that push-ups in the current academy are performed for a score. This score reaches a maximum when 50 repetitions are reached which could be providing a ceiling effect resulting in a lower score. These results show a trend that graduates from this academy possess a similar, if not higher, muscular endurance compared to other academies. When comparing push-ups to the general population, both males (53.79 ± 0.93 reps) and females (46.50 ± 1.68) in the 20-29 age range would be in the excellent category [23]. For ages ranging from 30 to 39, males (56.12 ± 2.26 reps) and females (49.42 ± 2.93 reps) would again be excellent [23]. Males (49.35 ± 1.43 reps) and females (50.00) aged 40–49 are also considered excellent when compared to the general population, though there was only one female in this age group.

These results provide evidence that while most domains of fitness increased, there appears to be a trend towards a more prominent increase in muscular endurance and aerobic capacity, a trend seen across other academy training programs [6]. While important aspects of fitness for a police officer, muscular strength and power are also vital to complete occupational tasks. Previous research has shown that measures of power and anaerobic fitness were more strongly correlated with police occupational tasks compared to measures of aerobic fitness [9]. The importance of these domains to occupational performance is reflected in the WSTB, itself a test of occupational simulations, with drills emphasizing strength and power. This trend is likely the result of physical training programs historically focused on muscular endurance and aerobic capacity, with physical training sessions containing bodyweight circuits and long-distance running [10,11]. In the future, academies should add elements of muscular strength and power to their physical training programs to better improve performance on work-simulated tasks and prepare their recruits for working in the field.

Recruits graduating from this academy tend to have a higher aerobic fitness level, and similar, if not higher, muscular endurance when compared to other academies. Compared to the general population, recruits from this agency have excellent muscular endurance as measured by the push-up. Worryingly, when comparing aerobic fitness, there is a trend of below-average scores when compared to the general population especially regarding the younger age ranges. Given the importance of aerobic fitness in both occupational task performance [2], and injury risk [6], this is something that should be addressed. Considering that previous research has shown that fitness decays over time upon graduation from the academy [16], strategies should be implemented aiming to further increase the aerobic fitness of recruits. It should be noted, however, that these findings contradict previous research in law enforcement which showsshow that officers are above age- related norms [29]. This may be a sign of changes in law enforcement fitness or may be specific to the population studied in this research. Future research will be necessary to clarify this point. As officers tend lose fitness throughout their careers [1], larger increases in aerobic fitness may allow officers to maintain higher levels of fitness as they age. Police departments may also be able to implement ongoing training programs to help maintain or even increase fitness in police officers after graduating from the academy.

There are certain limitations in this study thatwhich should be noted. As this is a retrospective study it may be limited by other confounding variables not addressed in this study, such as prior training history. Additionally, as the sample is drawn from one law enforcement agency, results from this study cannot be used as a representation of all law enforcement recruits. Another limitation of this

study is the lack of a true assessment of muscular power or muscular strength. While the occupational tasks performed utilize these qualities, a true assessment of these physiological capabilities would provide stronger evidence for the improvement, or lack thereof, of these qualities over the course of academy training. Lastly, the population in this study was weighted more towards younger individuals, specifically males. While representative of similar populations, increasing the number of female and older age subjects would allow for more robust comparisons.

## 5. Conclusions

Recruits were able to increase their fitness levels across almost all domains during an academy. However, a trend towards larger improvements in muscular endurance and aerobic capacity was seen compared to muscular strength. Even though larger improvements in aerobic capacity were seen, graduates from the academy still possessed lower aerobic fitness levels on average than the general population. While important, muscular endurance and aerobic capacity must not be solely focused on to the detriment of the other fitness domains. Adding a focus on muscular strength and power may improve performance on occupational simulated tasks, and possibly better prepare recruits for working as a law enforcement officers.

**Supplementary Materials:** The following are available online at http://www.mdpi.com/2071-1050/12/19/7944/s1.

**Author Contributions:** Conceptualization, D.J.M., B.S., and R.M.O.; methodology, D.J.M., B.S., and E.F.D.C.; software, D.J.M. and E.F.D.C.; validation, D.J.M., B.S., R.L., J.J.D. and R.M.O.; formal analysis, D.J.M., B.S., and E.F.D.C.; investigation, D.J.M. and B.S.; resources, R.M.O., R.L., J.J.D.; data curation, D.J.M., R.L., J.J.D.; writing—original draft preparation, D.J.M.; writing—review and editing, D.J.M., B.S., E.F.D.C., R.L., J.J.D. and R.M.O.; visualization, D.J.M., B.S., E.F.D.C., and R.M.O.; supervision, B.S., E.F.D.C., and R.M.O.; project administration, D.J.M., R.L., J.J.D., and R.M.O.; funding acquisition, D.J.M., R.L. and R.M.O. All authors have read and agreed to the published version of the manuscript.

**Funding:** This research was supported by an Australian Government Research Training Scholarship.

**Acknowledgments:** The authors would like to thank Lt. Joe Dulla and the rest of the members of the Los Angeles Sheriff's Department Recruit Training Center.

**Conflicts of Interest:** The authors have no conflict of interest to declare.

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
