# Peer review of "Developing the Fitness of Law Enforcement Recruits during Academy Training"

_sustainability, doi:10.3390/su12197944_

Round 1

Reviewer 1 Report

Dear Authors,

All comments you can find in the enclosed Word file.

Regards,

Rwviewer

Author Response

Thank you reviewer for taking the time to addreess this atricle. We hope the following changes improve the strength of this paper. 

1. This article is to be submiited for a special issue for Sustainability, Wellness Interventions for Sustainable Healthy Lifestyles Promotion in Tactical Populations, which has a focus on physical well-being and effectiveness of fitness programs. The introduction has been updated with information on fitness and its relation to long term physical and psychological health to strengthen Sustainability's message.  

2. Please note in the methods section, under statistical, statements have been added concerning age and gender, specifically how previous rearch in comparable populations have conducted similar analysis and found differnces. 

3. Please note the above response, and regarding the utility of looking at fitness between ages and gender. The statisical methods are appropriate for analysing these changes, and have been used previously. This previous use has been added to the methods section to strengthen the manuscript. 

4. The discussion section has been improved with comparisons to other studies made. In combination with the limitations section addressed below, this should increase the reflective nature of this section. 

5. Please note that a limitations section has been added to end of the discussion.

Thank you again for reviewing this article.  

Reviewer 2 Report

In the first place I would like to share the need to carry out work like the one you present.
They are necessary for the advancement of advanced society.

It is necessary to know the levels of physical condition of this group, before exercising, during their service and after their work.
For this reason I consider this work very important for the sector.

I would like to highlight the inclusion as a keyword of the words training and training, since it is constantly talked about at work. I would also like to see quotes from the authors of the different test-tests reflected in your work.
I think that to facilitate the reader's follow-up, they should include a descriptive table of the type of test and what it measures, since in the description of the different tests carried out, a lot of information may be exposed. Likewise, it is suggested to the actors the possibility of reviewing the manuscript regarding the inclusion or exclusion of the same to the subject who did not determine his sex, since if he is included it must be explained in which group his treatment was carried out. It should be reflected in the work who, how and when the control of said training was carried out and who supervised them. Like data collection. The limitations of the study should be reflected.

Author Response

Thank you for your feedback regarding this manuscript. Your contributions are appreciated.

Please see that physical training has been added to keywords to reflect your feedback. 

Secondly, regarding the table of physiological characteristics of tests. I am mindful of the number of tables that currently in this manuscript, however I do think this could be useful to some readers. Please see that a table has been added as a supplementary resource.

Please note that information outlining the participant who did not disclose their text has been added to the methods section. This participant was kept as part of the overall analysis but not factored into the analysis between sexes. 

Information regarding the physical training has been expanded in the methods section.

Please note that a limitations section has also been added at the end of the manuscript. 

Thank you again for your feedback. 

Reviewer 3 Report

Accept.

Author Response

Thank you for your feedback and agreement to accept this manuscript. Please note a thorough review of the manuscript was conducted to ensure proper language and grammer. 

Round 2

Reviewer 1 Report

Dear Authors,

After included changes and corrections, as well as knowing now the fact that this manuscript is addressed to the Special Issue, I recommend it to be published in present form.

Kind regards,

Reviewer